# Formic Acid Dehydrogenation over Ru- and Pd-Based Catalysts: Gas- vs. Liquid-Phase Reactions

**DOI:** 10.3390/ma16020472

**Published:** 2023-01-04

**Authors:** Estela Ruiz-López, María Ribota Peláez, María Blasco Ruz, María Isabel Domínguez Leal, Marcela Martínez Tejada, Svetlana Ivanova, Miguel Ángel Centeno

**Affiliations:** Departamento de Química Inorgánica e Instituto de Ciencia de Materiales de Sevilla, Centro Mixto CSIC-Universidad de Sevilla, Avda. Américo Vespucio 49, 41092 Sevilla, Spain

**Keywords:** hydrogen, formic acid, dehydrogenation, Ru catalyst, Pd catalyst, carbon nitride

## Abstract

Formic acid has recently been revealed to be an excellent hydrogen carrier, and interest in the development of efficient and selective catalysts towards its dehydrogenation has grown. This reaction has been widely explored using homogeneous catalysts; however, from a practical and scalable point of view, heterogeneous catalysts are usually preferred in industry. In this work, formic acid dehydrogenation reactions in both liquid- and vapor-phase conditions have been investigated using heterogeneous catalysts based on mono- or bimetallic Pd/Ru. In all of the explored conditions, the catalysts showed good catalytic activity and selectivity towards the dehydrogenation reaction, avoiding the formation of undesired CO.

## 1. Introduction

The energy crisis in which we are all involved can only be solved by decreasing the global energy demand and restricting the use of non-renewable, traditional energy sources. Due to the fact that there are still about 770 million people without access to electricity [1], it seems unlikely that we will achieve a decrease in the global demand. Fortunately, most environmental politics are currently focused on the search for clean, green, and totally renewable energies in which hydrogen appears as a major player. In particular, green hydrogen (produced from low-carbon, renewable sources) is considered as key element to aid in the decarbonization of the current energy model since its combustion generates CO_2_-free energy. Notwithstanding, its unsolved transport and storage issues retard its launch and implementation as an energy vector [2].

Among the main solutions to hydrogen transport and storage issues, liquid organic hydrogen carriers (LOHCs) have emerged as one of the most promising and attractive materials for hydrogen storage since they are compounds that are able to capture and release hydrogen through chemical reactions. Their ability to generate in situ hydrogen in conjunction with their high gravimetric storage density (2–4 kWh kg^−1^) as compared to metal hydrides (<1 kWh kg^−1^) or compressed hydrogen gas (2 kWh kg^−1^) has converted these materials into a safer option for energy storage via hydrogen [3,4]. Furthermore, the current crude-oil-based infrastructure could serve for the implementation of LOHCs since they are liquid at ambient conditions and present properties similar to those of traditional liquid oils [3].

Formic acid (FA) unites most of the required features to be considered as an appealing LOHC since it possesses a proper hydrogen weight (4.4 wt.%) and volumetric capacity (53 g_H2_ L^−1^) [5], as well as kinetically stable properties that help its handling and transportation (therefore, the current infrastructure could be used). FA also presents low toxicity and flammability at ambient conditions, and its synthesis and dehydrogenation can be performed under mild conditions [6]. Furthermore, and most importantly, it can be produced from renewable sources. Despite its vast current industrial production (~80%) involving the carbonylation of methanol and further methyl formate hydrolysis [4,7,8,9], FA can be also obtained from CO_2_ capture and hydrogenation or from different biomass feedstocks, such as glucose, glycerol, lignin, or sugar oxidation [8,9,10,11].

Hydrogen production from formic acid takes place via formic acid decomposition (FAD), in which two thermodynamically stable reactions are involved: the dehydrogenation reaction (Equation (1)), where H_2_ is produced along with CO_2,_ and the formic acid dehydration reaction (Equation (2)) to produce CO and H_2_O.
(1)$$\mathrm{HCOOH}\;\to {\;H}_{2}{+\mathrm{CO}}_{2}$$
(2)$$\mathrm{HCOOH}\;\to {\;H}_{2}O+\mathrm{CO}$$

Taking into account that any LOHC reaction must feed a fuel cell (FC) for efficient and clean hydrogen utilization, their extremely low CO tolerance must be taken into consideration when treating FA as hydrogen carrier [12,13]. Proton-exchange membrane fuel cells (PEMFCs) and their Pt catalysts are quite sensitive to CO poisoning (15 ppm of CO in the fuel gas could result in a 30% current loss [14]) since the latter strongly bonds to Pt and hinders hydrogen adsorption. Despite the unceasing effort made to enhance CO tolerance, compositions higher than 3% could not be accepted in the most favorable cases, using phosphoric-acid-doped polybenzimidazole membranes in high temperature PEMFCs [13,15,16]. In any case, the CO presence in the gas fed to FCs would reduce the fuel cells’ performance and durability.

Considering the above, a complete selectivity to formic acid dehydrogenation instead of an FA dehydration reaction is one of the main goals for achieving an important level of effectivity in the FAD systems. Liquid—aqueous-phase FAD would help to suppress dehydration reaction, and it has been studied extensively [17,18,19,20,21] although many of these works used homogeneous catalysts that limit the large-scale application of the process [22]. Regarding gas-phase FAD, it has also been studied [23,24,25,26], and it has been found that the addition of steam could shift the selectivity towards the dehydrogenation reaction [27]. Comparing both the liquid- and gas-phase reactions, it seems that the latter may be more attractive from an industrial point of view. Although the reaction conditions for the liquid phase are more favorable (FAD has even been achieved at room temperature [28]), the continuous-flow reactor design typically used in the gas phase easily allows for the continuous and stable production of hydrogen, which is almost impossible to achieve while using a semi-batch reactor in the liquid phase. Moreover, the recovery, regeneration, and reusability of the gas-phase catalyst is more favorable than that of the liquid-phase catalyst, which can suffer some deactivation.

Carbon- and carbon-nitride (C_3_N_4_)-based materials have been used as supports for heterogeneous catalysts in both the liquid and gas phases due to their high thermal and chemical stability, low price, and high availability. Due to their aromatic C-N heterocycles, they are thermally stable, even in air up to 600 °C, and they are chemically stable in most solvents because of strong interlayer van der Waals interactions, which provide the C_3_N_4_ with a high specific surface area. Additionally, their composition (using abundant elements such as C, N, and H) not only assures their easy and cheap preparation from different sources, but it also provides the ability to tune their composition, and hence their structural properties [29,30,31].

As for the active phase, metals such as Au, Ag, Pd, Pt, Ru, and Ir have been widely studied for the FAD reaction [19,28,32,33,34,35], with Pd being the preferred one due to its high stability and selectivity [23]. In fact, the current tendency to improve FAD performance consists of the application of bi- or tri-metallic Pd-based catalysts (as alloys, core–shell structures, etc.), with the aim of modifying the catalytic Pd NPs surface to achieve higher activities and selectivities [17,19,28,33]. Moreover, Pd-based NPs supported on N-doped carbon have proven to be excellent catalysts for several organic reactions [36,37,38]. On the other hand, Ru has mainly been used as a homogeneous catalyst [29] although interesting results for gas-phase FAD have been found while supported on metal and covalent organic framework (MOF and COF) materials [32,39].

To all of that discussed above, this work provides an attempt to add to the study of FAD behavior a series of experiments with mono- and bimetallic Pd/Ru catalysts supported on graphitic C_3_N_4_. Their activity was evaluated in both the liquid and gas phase, with a final aim of achieving a maximum conversion and selectivity towards H_2_, inhibiting CO production, and to be able to produce a stable and clean hydrogen stream.

## 2. Experimental

### 2.1. Catalysts and Chemicals

For this study, three different catalysts (monometallic Pd, monometallic Ru, and bimetallic PdRu, all supported on carbon nitride, C_3_N_4_), were synthesized. The used support, C_3_N_4_, was obtained after calcination of commercial melamine (Sigma-Aldrich^®^) at 650 °C for 2 h (2 °C min^−1^ heating rate) in a capped crucible. The chemical precursors for Pd and Ru were palladium nitrate and ruthenium (III) nitrosyl nitrate solution, both purchased from Johnson Matthey^®^. These precursors were deposited via wetness impregnation on the prepared support, targeting 5 wt.% metal loading in each catalyst (with a Pd:Ru 1:1 molar ratio in the case of the bimetallic catalyst). The metal charge was selected to be high enough for an important liquid-phase hydrogen production, as studied previously [40]. The catalysts were labelled as Pd/C_3_N_4_, Ru/C_3_N_4_, and PdRu/C_3_N_4_. Prior activity measurements, the catalysts were treated thermally at 250 °C for 1 h in an inert atmosphere (N_2_, 100 mL·min^−1^) and then reduced at 300 °C for 1 h (N_2_/H_2_, 1:1, total flow = 100 mL·min^−1^). 

### 2.2. Characterization Methods

Elemental analyses were performed on an Elemental Analyzer LECO TruSpec CHN. XRD measurements were performed on an X’Pert Pro PANalytical diffractometer equipped with a Cu anode and working at 45 kV/40 mA. The diffractograms were recorded from 10 to 90° 2θ with a 0.05° step size and a 300 s step time. The structure/phase determination was performed by comparison with the Crystallography Open Database (COD) using X’Pert Highscore Plus software. Average Pd and Ru crystallite sizes were calculated using the Scherrer equation over the most intense diffractions (Pd(111) and Ru(101), respectively).

ICP-OES was used to determine and measure the real metal loading achieved with each catalyst using an ULTIMA 2 Spectro ICP spectrometer. Prior to performing the analyses, 5 mg of catalyst was added to a 3 mL HCL + 2 mL HNO_3_ + 2 mL H_2_O_2_ solution and then placed in a microwave oven to thermally treat it for 90 min (heating up to 230 °C, 15 min at 230 °C, and cooling down to ambient temperature). Finally, it was diluted with distilled water up to 50 mL.

TEM micrographs were acquired using an FEI Talos electron microscope equipped with a field emission filament operating at 200 kV. Digital images were taken with a side-mounted Ceta 16M camera. A few milligrams of the sample were deposited directly onto a 200 mesh holey carbon-coated copper TEM-grid and introduced to the microscope. Based on the TEM micrographs, and following Equation (3), the mean particle size of each catalyst was calculated by counting around 200 particles.
(3)$$D_{p}=\frac{\sum n_{i}d_{i}^{3}}{\sum n_{i}d_{i}^{2}}$$

### 2.3. Catalytic Set-Up

Two different catalytic set-ups were used to perform the liquid- and gas-phase reactions.

A four-neck round glass semi-batch reactor (250 mL) was the main component of the liquid-phase set-up. This reactor was continuously flushed with N_2_ (100 mL min^−1^) inlet/outlet, either to purge the system or to act as a carrier. This stream was also used as an internal pattern for the gas chromatograph calibration and thus the reaction evaluation. A cooling system was connected to the outlet stream. This set-up has been described in detail in previous studies [40]. The experimental procedure was as follows: 100 mL of 1M formic acid aqueous solution (formic acid: Sigma-Aldrich^®^, ACS reagent > 98%) was added to the reactor and continuously stirred (1036 rpm) as the temperature increased up to 60 °C. The high stirring rate was chosen to diminish in principle any possible problems in hydrogen transfer from the liquid to the gas phase. At that time, 0.1 g of the corresponding catalyst (300–400 μm grain size) was added to the reactor, pointing at the beginning of the reaction. The stirring was continued during the reaction in order to avoid or minimize the possible diffusional problems. Moreover, ammonium formate (Alfa Aesar^®^) was also used as additive for the aqueous solution in some tests.

A gas chromatograph (490 Micro GC System, Agilent^®^; column: Molecular Sieve 5A) coupled to a CO_2_ infrared sensor (Vaisala, MI70) were used to measure the obtained gas products. These analytic systems allow for the measurement of H_2_, CO, CO_2_, CH_4_, and other hydrocarbons. Turnover number (TON) and turnover frequency (TOF) were evaluated at t = 120 min and calculated following Equations (4) and (5), respectively.
(4)$$\mathrm{TON}=\frac{{\mathrm{mmol}\;\mathrm{of}\;H}_{2}\;\mathrm{produced}}{\mathrm{mmol}\;\mathrm{of}\;\mathrm{Pd}}$$
(5)$$\mathrm{TOF}\;\left(h^{-1}\right)=\frac{{\mathrm{mmol}\;\mathrm{of}\;H}_{2}\;\mathrm{produced}}{\mathrm{mmol}\;\mathrm{of}\;\mathrm{Pd}\cdot \mathrm{time}\;(h)}$$

A fixed-bed stainless-steel reactor (250 mm in length, 9 mm in internal diameter) was used for the gas-phase reaction, fed with a pre-heated inlet stream, using a syringe pump, an evaporator, and a mixer to homogenize the reaction flow. The fixed-bed consisted of 0.5 mL of the thermally treated and reduced catalyst with a 300–400 μm grain size. A heat exchanger was used to condense the outlet liquid phase (water and non-reacted formic acid), and the gas phase (H_2_, CO, CO_2_, and CH_4_) was continuously monitored by an ABB AO2020 analyzer. The gas-phase set-up is schematized in Figure 1.

Silicon carbide (SiC, Alfa Aesar^®^, 300–425 μm grain size) was used as the blank reaction. A 100 mL·min^−1^ (5% *v*/*v* formic acid, 25% *v*/*v* distilled water and 70% *v*/*v* N_2_) flow fed the reactor, and the gas hourly space velocity (GHSV) was about 18,000 h^−1^. Two different experiments were performed with the gas-phase set-up. First, a temperature screening from 150 to 400 °C (25 °C/step, 40 min/step) was carried out, obtaining steady-state gas production at each temperature. Then, at the selected temperature of 250 °C, a long-term experiment was performed for 30 h in order to test the catalyst’s stability. The formic acid conversion, the product’s selectivity, and the hydrogen yield were calculated following Equations (6)–(8):(6)$${\mathrm{Formic}\;\mathrm{acid}\;\mathrm{conversion},\;x}_{\mathrm{FA}}\;\left(\%\right)=\frac{n_{{\mathrm{CO}}_{2}}{+n}_{\mathrm{CO}}{+n}_{{\mathrm{CH}}_{4}}}{n_{\mathrm{FA}}^{0}}\cdot 100$$
(7)$${\mathrm{Selectivity},\;s}_{i}\;\left(\%\right)=\frac{n_{i}}{n_{H_{2}}{+n}_{{\mathrm{CO}}_{2}}{+n}_{\mathrm{CO}}{+n}_{{\mathrm{CH}}_{4}}}\cdot 100$$
(8)$${\mathrm{Hydrogen}\;\mathrm{yield},\;y}_{H_{2}}\left(\%\right)=\frac{n_{H_{2}}}{n_{H_{2},\;\mathrm{theoretical}}}\cdot 100$$
where n^0^_FA_ is the FA molar flow fed to the reactor, n_i_ is the obtained molar flow for the corresponding species, and n_H2_, theoretical corresponds to the theoretical maximum molar flow of the obtained H_2_ following Equation (1) stoichiometry. The formic acid conversion was also checked by HPLC recovering condensate at each temperature after the reactor (column Hi-Plex H, milliQ water as mobile phase).

## 3. Results and Discussions

The different characterization techniques were performed in order to corroborate the metal loading as well as observe the catalysts’ structure, particle size, and distribution.

The metal loading obtained via ICP-OES analysis matched the intended experimental values within a ±0.3 range (shown in Table 1). Elemental analysis on the C_3_N_4_ support indicated the presence of some hydrogen remaining after the melamine thermal treatment, with the final atomic composition of the support being C_3_N_4.37_H_1.85_.

The XRD patterns of reduced C_3_N_4_, Pd/C_3_N_4_, Ru/C_3_N_4_, and PdRu/C_3_N_4_ are displayed in Figure 2. The diffractions observed in all patterns at 13° and 27.6° are characteristics of the lattice (100) and (002) planes of carbon nitride [41], both attributed conventionally to the graphitic stacking of the C_3_N_4_ structure. Whereas the former is indicative of an in-plane repeating unit (interplanar distance of 0.675 nm), the stronger (002) diffraction at 27.6° corresponds to a period of 0.326 nm due to the layered stacking characteristic of conjugated aromatic systems [42,43]. These two characteristic peaks remained unaltered in all samples, dismissing the possibility of an insertion of Pd or Ru species at the interlayer [44,45]. The patterns obtained for the three catalysts have been compared to standard Pd (COD, ref. 96-900-8479) and Ru (COD, ref. 96-900-8514) (both marked by dotted lines in the figure).

Regarding the Ru-containing samples for both the monometallic and bimetallic catalysts, the diffractions matched the characteristic lattice planes of the hexagonal Ru crystal structure. The face-centered cubic Pd^0^ is present in the monometallic catalyst, whereas for the bimetallic sample, its presence is hard to confirm, suggesting a very good dispersion, but not an alloy formation. On the other hand, for the monometallic Pd catalyst, a splitting of the main Pd diffraction peaks can be appreciated. It has been previously reported that hydrogen atoms are able to diffuse into the Pd lattice, leading to its expansion [46,47]. The diffusion is actually detectable in the XRD patterns since it provokes a shift towards lower 2θ values (COD, ref. 96-900-8698), as can be appreciated in Figure 2b. The double peaks observed for all diffractions indicate the presence of both Pd(0) and PdH_x_ or H-loaded Pd species. The origin of this H diffusion resides either in the reduction step during the catalyst synthesis or in the remaining hydrogen from the melamine calcination process. The average Pd and Ru crystallite sizes (calculated using the Scherrer equation over Pd(111) or Ru(101)) are shown in Table 1.

HR-TEM was used to calculate precisely the metal particle size as well as its distribution (Figure 3 and Table 1). Comparing the monometallic catalysts, bigger particle sizes were found for the Ru catalyst, whereas the bimetallic catalysts exhibited a medium size; that is to say, the presence of Pd seems to diminish the Ru particle size. The differences in size detected by XRD and TEM are not unexpected, taking into account the errors that can occur in the average crystallite size evaluation, especially for the doubled-peak Pd sample and the limited possibility of detecting very small particles (XRD limit of detection < 3 nm). A monomodal TEM distribution was found for all catalysts, with an average size variation between 2.6 and 4.2 nm.

### 3.1. Liquid-Phase FA Dehydrogenation

First, the catalytic activity was studied in liquid-phase conditions. In all cases, only H_2_ and CO_2_ were detected as products, whereas no traces of CO, CH_4_, or other hydrocarbons were detected. In other words, the selectivity was completely shifted towards the desired dehydrogenation reaction. The cumulative volume of produced hydrogen, the total produced gas (H_2_ + CO_2_), and the H_2_/CO_2_ molar ratio are shown in Figure 4.

Evaluating the results of the hydrogen production for the three catalysts, one can conclude that Pd is the only metal that acts as an active phase. The monometallic Pd catalyst reached values of 140 mL of H_2_ in 120 min, while the monometallic Ru did not show activity in the reaction. In the same way, the presence of Ru in the bimetallic catalyst lowered the activity of the catalyst to the production of about 60 mL of H_2_ under the same conditions. However, comparing the Pd/C_3_N_4_ and PdRu/C_3_N_4_ catalytic performances based on TON and TOF, it is observed that the obtained values are rather similar (Table 2), considering only Pd as an active phase. Regarding the H_2_/CO_2_ molar ratio, values from the monometallic Ru catalysts were not calculated since they did not show catalytic activity. In the other two catalysts, after 20 min of reaction, values were close to 1, as expected due to the reaction stoichiometry (Equation (1)). During the first 20 min of the reaction, the high ratio values can be explained since the reaction was just starting, and CO_2_ and H_2_ are measured by different devices. We must not forget that the support can also play an important role through its interaction with the metal [48]. One can speculate that the nitrogen species located on the C_3_N_4_ surface are able to play a dual role: they stabilize the Pd particles and provide adsorption sites, as well as reducing the electron density on the Pd surface, thus allowing for easier adsorption of reactives [48,49,50].

In contrast, and compared with Pd, Ru behaved so differently in the liquid-phase FAD reaction. Although frequently employed as homogeneous catalysts, the Ru complexes do not seem active while supported in C_3_N_4_ [29] in the liquid phase due to the competitive absorption of water over the metal sites and surface hydroxylation, thus making difficult the arrival of the formic acid to the active site. On the contrary, the Pd catalyst appears to be very active in the FAD reaction. It is believed that the main reaction path in this case is through the adsorption of an intermediate carboxyl on Pd (111), with FA acting as a precursor for it. The FA (weakly adsorbed) is converted to a carboxyl, which suffers O-H bond cleavage and generates H_2_ and CO_2_. Notwithstanding, the intermediate carboxyl could also break the C-O bond and hence produce CO and H_2_O [51]. The latter is hardly possible due to the important hydroxylation of the surface in aqueous media, making possible only the first mechanism. Although resulting in lower total hydrogen production, the bimetallic catalyst actually benefits from the presence of Ru, facilitating the carboxyl formation on Pd sites and generating a similar TOF as the monometallic Pd.

Pd and Pd-Ru catalysts were also tested in a formic acid: ammonium formate mixed solution (FA:AF 1:9 molar ratio) (Figure 5).

The presence of additives such as formate led to an increase in the reaction rate since its electron-donation ability towards the Pd surface could induce its favorable adsorption and fast dehydrogenation, shifting the formic acid/formate equilibrium towards formate production. In this scenario, the formate ion acts as an active intermediate (formate ion, HCOO^−^ binds first to Pd particles), and at a certain concentration, it promotes a liquid-phase FAD reaction [48,51,52]. What is more, in FA:AF aqueous solution, and according to Equation (9), NH_3_ is present. It has been reported that the addition of amine or the modification of the support with amine could enhance the FAD catalytic activity [53]. As shown in Figure 5, and in comparison with the results presented in Figure 4 (without the AF additive), hydrogen production was enhanced more than threefold.
(9)$${\mathrm{HCOONH}}_{4}{+H}_{2}O\;⇄{\;\mathrm{HCOOH}+\mathrm{NH}}_{3}{\cdot H}_{2}O$$

For comparison, TON and TOF values (calculated via Equations (4) and (5), respectively) are summarized in Table 2. Despite the complexity of comparing different catalysts tested in different conditions, it could be concluded that the activity of these catalysts is in line with currently published results.
materials-16-00472-t002_Table 2Table 2TON and TOF values for different Pd/C_3_N_4_ catalysts.CatalystReaction Conditions(Catalyst Weight (mg), Reactant Mixture, Temperature (°C) and Time (min))TON ^1^TOF ^1^ (h^−1^)Ref.Pd/C_3_N_4_ 5%100 mg, FA 1M, 60 °C, 200 min73.5936.80This workPd/C_3_N_4_ 5%100 mg, FA:AF ^2^ 1M (1:9), 60 °C, 200 min259.07129.54This workPdRu/C_3_N_4_ 5%100 mg, FA 1M, 60 °C, 200 min60.6830.34This workPdRu/C_3_N_4_ 5%100 mg, FA:AF ^2^ 1M (1:9), 60 °C, 200 min190.4795.24This workPd/gC_3_N_4_ 1.1%100 mg, FA:SF ^2^ 6M (1:9), 25 °C, 120 min 383.12191.56[48]Pd/mpg-C_3_N_4_ 3.2%40 mg, SF ^2^ 4M, 60 °C, 120 min519.63259.81[49]Pd/mpg-C_3_N_4_ 9.5%50 mg, FA 1M, 25 °C, 180 min92.5246.26[50]Pd/C 10%100 mg, FA 1.33M, 60 °C, 300 min178.1689.08[54]Pd/C 2.3%55 mg, FA:SF ^2^ 1.2M (1:1), 25 °C, 150 min112.6756.33[55]Pd/201 (resin) 10%50 mg, FA 0.25M, 50 °C, 400 min9.504.75[56]^1^ TON and TOF were calculated at 120 min in all cases. ^2^ AF refers to ammonium formate, and SF to sodium formate.

### 3.2. Gas-Phase FA Dehydrogenation

Gas-phase FAD activity in terms of H_2_, CO_2_, CO, and CH_4_ volumetric flows vs. temperature is shown in Figure 6. As observed from the blank experiment with SiC, formic acid thermal decomposition starts at 275 °C and reaches a complete conversion at temperatures above 350 °C.

Stable and steady hydrogen production can be observed in the 150–350 °C range for the monometallic Pd. Hydrogen is produced at temperatures as low as 150 °C, much earlier than the thermal FAD observed. The conversion values oscillated between 90 and 100%. As for the liquid-phase dehydrogenation, Pd-based catalysts demonstrated a better performance than other metal-based catalysts [57]. What is more, they showed very high selectivity towards the FAD reaction, as confirmed by the negligible production of CO at low temperatures. As expected, the water added to the system favors dehydrogenation via the Le Chatellier principle, but also the water–gas shift reaction (${\mathrm{CO}+H}_{2}O\;\to {\;\mathrm{CO}}_{2}{+H}_{2}$) occurred at low temperatures and hence the conversion of possible CO to CO_2_. Indeed, Solymosi et al. [58] found that pure H_2_ cannot be obtained through formic acid decomposition in the absence of water at temperatures above 50 °C. At higher temperatures (above 300 °C), the CO production noticeably increased in the case of the monometallic Pd catalyst, being that the water–gas shift reaction was unfavorable at these temperatures, even more so, considering the strong presence of CO_2_ and H_2_, which should shift the equilibrium towards the reverse water–gas shift reaction (${\mathrm{CO}}_{2}{+H}_{2}\to {\;\mathrm{CO}+H}_{2}O$).

Concerning the monometallic Ru catalyst, a different behavior was found. The catalyst shows some CO production at low temperatures (<275 °C) due to the formic acid dehydration reaction (Equation (2)), obtaining CO and H_2_O as products. According to previous studies [59,60], Ru catalysts do not show activity for the water–gas shift reaction at temperatures below 350 °C. Then, at low temperatures, and since this reaction is not favored, the CO obtained via the dehydration reaction is not further reacted. However, a change of selectivity occurs at temperatures above 275 °C, when CO is no longer produced, giving way to a CH_4_ production. This selectivity shift could be explained through the CO methanation reaction (${\mathrm{CO}+3H}_{2}\;\to {\;\mathrm{CH}}_{4}{+H}_{2}O$), which would also explain the hydrogen consumption (“hydrogen decrease in yield”) observed in that temperature range. Ruthenium has proven to be an excellent catalyst for the CO methanation reaction, regardless of the support used [61,62,63]. It is even effective for selective CO methanation in H_2_-rich gas streams under a low CO concentration and in the presence of CO_2_ and water in a temperature window rather similar to the temperature range used in the present study [64]. Whereas the feed of a CO/CO_2_ mixture would favor the reverse water–gas shift reaction, the presence of water inhibits it [62] and favors CO methanation. Moreover, it has been reported that the presence of water does not affect the latter reaction in some cases [65], or even helps in others [62]. As for the CO_2_ hydrogenation (${\mathrm{CO}}_{2}{+4H}_{2}\;\to {\;\mathrm{CH}}_{4}{+2H}_{2}O$), water vapor did not showcase a clear role since it was observed that it could not affect [62], or shift the reaction towards higher temperatures [65], or even completely inhibit the reaction [66]. In our case, the monometallic Ru catalyst achieved a complete conversion of CO to CH_4_ at temperatures above 250 °C, with the CO_2_ not involved (via the Sabatier reaction) due to the observed H_2_ flow decrease following the stoichiometry of the CO methanation. The role of CO_2_ seems irrelevant in CH_4_ formation, thus its evolution with the temperature is completely linked to the selectivity towards formic acid dehydrogenation or dehydration coupled with the water–gas shift reaction.

Compared to Ru, our Pd catalyst was able to convert CO produced at high temperatures in neither CO_2_ nor in CH_4_. Pd has been found to be practically inactive for the CO methanation reaction, presenting poor activities (CO conversions < 10%) at temperatures below 400 °C, and achieving only a 22% CO conversion at temperatures up to 550 °C, as reported in the literature [64,65]. It has also been reported that over Pd catalysts, the CO conversion remains unaffected by the presence of water, with suppressed CH_4_ selectivity [65].

The result of the combination of both mechanisms can be clearly observed in the bimetallic catalyst, where the CO formation was similar to that observed for the monometallic Pd catalyst at low temperatures and similar to Ru catalysts at a high temperature. The WGS reaction present at low temperatures remains unfavorable above 300 °C, where the increased CO production was rapidly switched to methane via a CO hydrogenation reaction. The bimetallic catalyst appears to compel the action of both metals.

H_2_ and CO selectivity and hydrogen yields (empty symbols) are summarized in Figure 7. For none of the catalysts, did the CO selectivity surpass 6%, and for the Pd and Pd-Ru catalysts, a CO-free gas stream was obtained, but in different temperature ranges. For the latter, the selectivity towards formic acid dehydrogenation, and as a consequence H_2_ production, was around 100%. In terms of H_2_ yields, it can be observed that values close to 100% were obtained at most temperatures.

In previous studies of the FAD reaction, Pd- and Ru-based catalysts were incapable of achieving complete formic acid conversion and 100% H_2_ selectivity at once [23,39,58]. Arzac et al. [23] prepared a Pd-C thin film supported on a SiC monolith, giving a conversion and selectivity of 80% and 88% at 350 °C, respectively. Selectivities higher than 90% were found at temperatures below 250 °C, with conversion values nearing 20% in dry conditions. Solymosi et al. [58] also tested Pd and Ru carbon-supported catalysts, giving a total conversion at 250 °C, with hydrogen selectivity around 90% in dry conditions, which improved with the addition of water. Notwithstanding, their best values of H_2_ yields were close to 92% in the case of the Pd catalyst and close to 63% in the case of the Ru catalyst, both at 200 °C.

A stability test was also performed on each catalyst for 30 h at 250 °C (Figure 8). This temperature was selected in order to compare the real stability since activity, conversion, and selectivity were similar for all of them (as observed in Figure 4 and Figure 5).

The three catalysts manifested a stable performance, achieving conversion values higher than 90% without decreasing over time. The cumulative volume of the obtained hydrogen presented a linear increase, corroborating the continuous production without signs of catalytic deactivation. After this stability test, XRD analysis was performed on the used catalysts (Figure 9).

According to the XRD patterns, all catalysts remained unaltered after the stability tests. Nevertheless, it was noticeable that, for the spent monometallic Pd catalyst, the double peaks of the Pd/PdHx system disappeared, with only the H-loaded species produced by hydrogen diffusion into the remaining Pd(0) lattice present.

## 4. Conclusions

C_3_N_4_ has been proven to be an effective and stable support for Pd-based catalysts for FAD reactions in both liquid- and gas-phase reactions, showing a complete selectivity towards the desired dehydrogenation reaction. The Ru catalyst is inactive in the liquid-phase reaction, but its presence is welcomed in the bimetallic catalyst for the synergic effect that it causes, with Pd particle size and activity showing the same TOF as that of the monometallic Pd catalyst. Nevertheless, the total hydrogen production is superior for the monometallic Pd catalyst. As expected, the addition of ammonium formate in the liquid-phase conditions tripled the catalytic activity of both Pd and PdRu/C_3_N_4_, possibly due to the presence of both formate ions and NH_3_. As for the gas-phase reaction, no matter the active phase, the samples showed total formic acid conversion in the whole range of temperatures, suggesting an excess of active centers. Once again, monometallic Ru seems to be the different sample, shifting the selectivity towards CO or CH_4_ formation, depending on the temperature. The catalysts present stable performance (conversion remained at values higher than 90% for 30 h of performance) and a H_2_ yield close to 100% in all temperature ranges in gas-phase conditions. Both reactions (gas- and liquid-phase) produce CO-free hydrogen, but it is the gas-phase reaction which allows for continuous, stable production. What is more, it offers the possibility of reducing the active phase.

## Figures and Tables

**Figure 1 materials-16-00472-f001:**
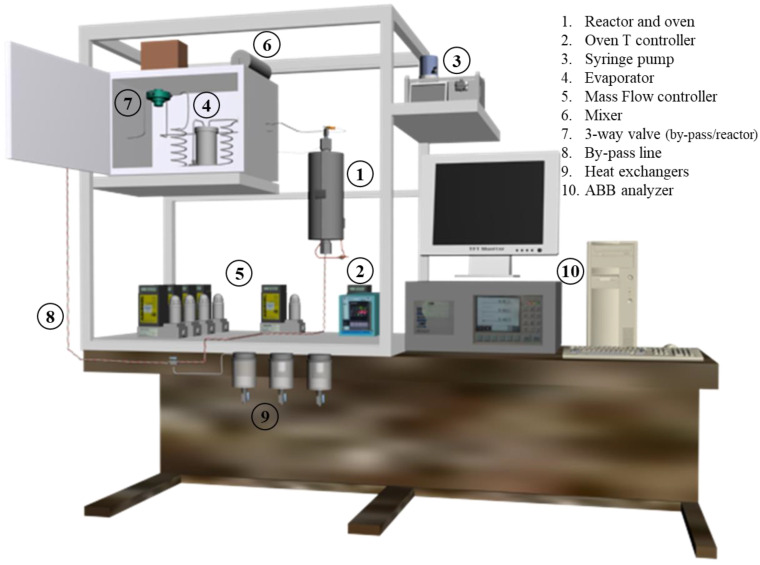
Gas-phase reaction set-up scheme.

**Figure 2 materials-16-00472-f002:**
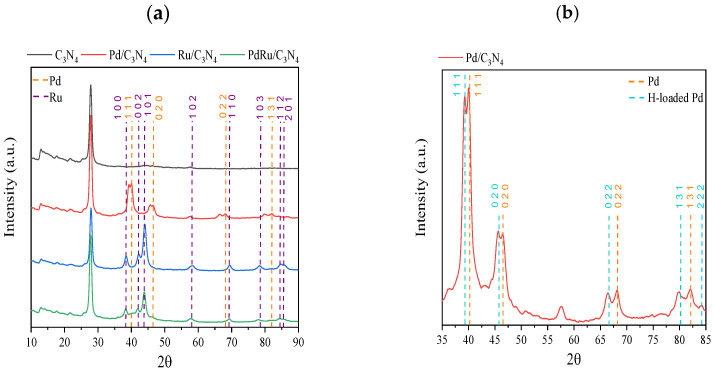
(**a**) XRD patterns of C_3_N_4_, Pd/C_3_N_4_, Ru/C_3_N_4_, and PdRu/C_3_N_4_; (**b**) Zoom of XRD pattern of Pd/C_3_N_4_ catalyst.

**Figure 3 materials-16-00472-f003:**
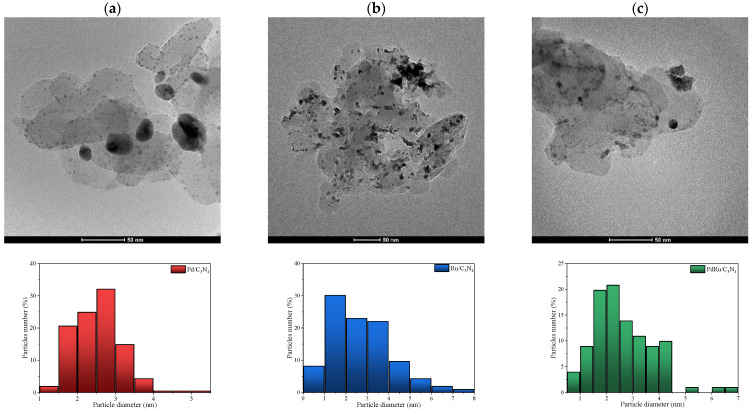
HR-TEM images obtained for (**a**) Pd/C_3_N_4_, (**b**) Ru/C_3_N_4,_ and (**c**) PdRu/C_3_N_4_, and the corresponding size-distribution histograms.

**Figure 4 materials-16-00472-f004:**
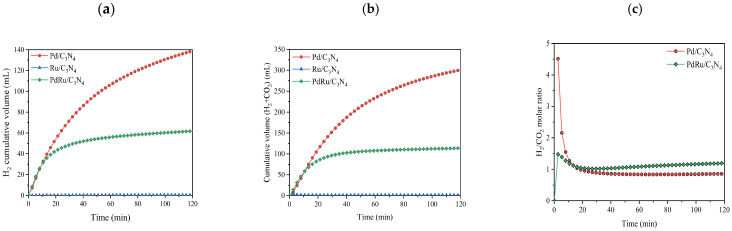
Cumulative volume of (**a**) hydrogen produced; (**b**) total gas produced in liquid-phase conditions for the FAD reaction (1 M FA. T = 60 °C); (**c**) H_2_/CO_2_ molar ratio.

**Figure 5 materials-16-00472-f005:**
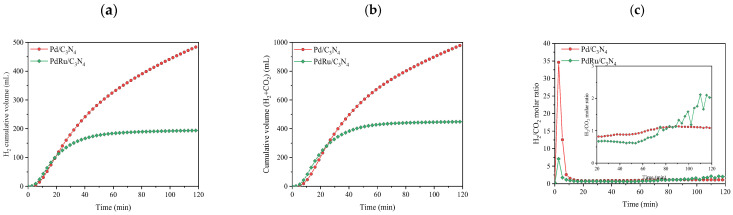
Cumulative volume of (**a**) hydrogen produced and (**b**) total gas produced in liquid-phase conditions for the FAD reaction (1 M FA:AF (1:9 molar ratio). T = 60 °C); (**c**) H_2_/CO_2_ molar ratio.

**Figure 6 materials-16-00472-f006:**
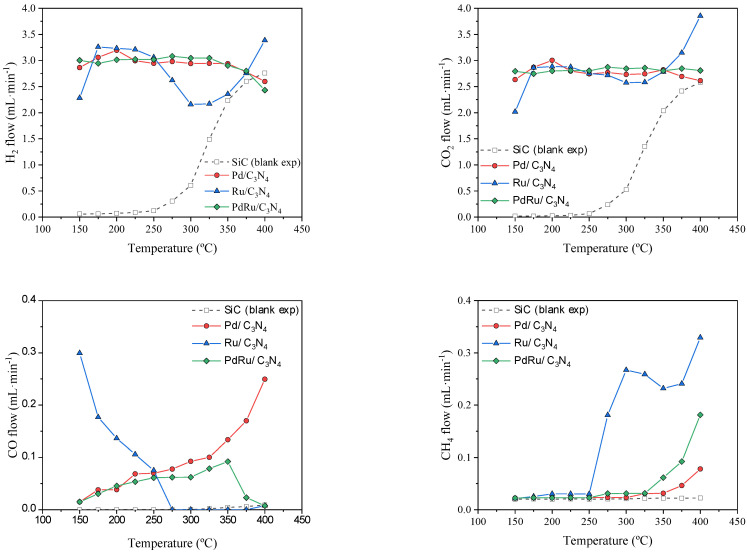
FAD reaction activity: H_2_, CO_2_, CO, and CH_4_ volumetric flows vs. temperature.

**Figure 7 materials-16-00472-f007:**
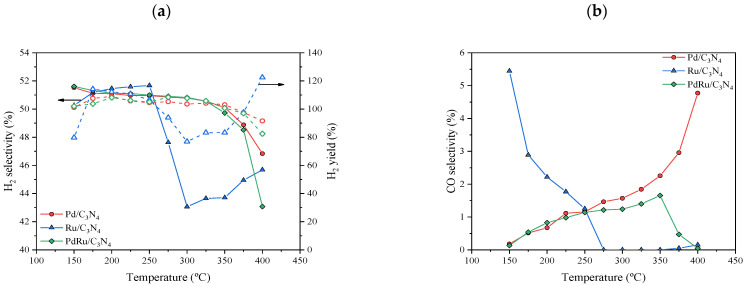
(**a**) H_2_ selectivity (full symbols) and H_2_ yields (empty symbols), and (**b**) CO selectivity.

**Figure 8 materials-16-00472-f008:**
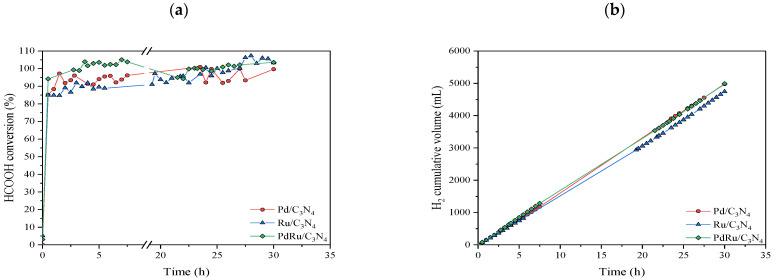
Stability test at 250 °C: (**a**) Formic acid conversion vs. time; (**b**) Cumulative volume of obtained hydrogen.

**Figure 9 materials-16-00472-f009:**
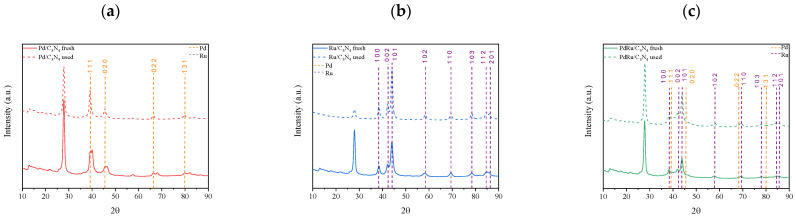
Comparison of XRD patterns of the three fresh and used catalysts: (**a**) monometallic Pd, (**b**) monometallic Ru, and (**c**) bimetallic Pd-Ru catalysts.

**Table 1 materials-16-00472-t001:** Metal loading obtained via ICP-OES analysis, crystallite size calculated via Scherrer’s equation from XRD patterns for each catalyst, and mean particle size calculated by HR-TEM.

Catalyst	Metal Loading (wt.%)	Crystallite Size (XRD, nm)	Mean Particle Size (TEM, nm)
Pd/C_3_N_4_	4.8	17.2 (Pd)	2.8
Ru/C_3_N_4_	4.7	15.3 (Ru)	4.2
PdRu/C_3_N_4_	2.6 (Pd) 2.2 (Ru)	9.8 (Ru)	3.6

## Data Availability

Data available upon request.

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
