# Peer review of "Formic Acid Dehydrogenation over Ru- and Pd-Based Catalysts: Gas- vs. Liquid-Phase Reactions"

_materials, 2023, doi:10.3390/ma16020472_

Round 1

Reviewer 1 Report

The paper reported a simple method to deposit Ru and Pd bimetal nanoparticles on graphitic C3N4. The as-prepared PdRu/C3N4 catalyst exhibited good catalytic activity and selectivity towards the dehydrogenation reaction in both, liquid and gas phase. Therefore, this manuscript is recommended for publication after the following issues are properly addressed.

1.      The author claimed that the Pd-Ru bimetal was formed. More detailed characterization (e.g. high-resolution TEM and EDS mapping) should be provided.

2.      The exact metal loading of Pd and Ru in PdRu/C3N4 catalyst should to be elaborated.

3.      The particle distribution of reused catalyst is recommended to be analyzed by TEM and compared with the freshly prepared one.

4.      Some related works about Pd-based bimetallic catalysts is recommended to be cited (not exclusive): J. Catal. 2022, 413, 779-785.; Chem. Eng. Sci. 2021, 231, 116303; ACS Appl. Nano Mater. 2021, 4, 5854−5863.; Appl. Surf. Sci., 2019, 489, 477-484; Carbon Lett. (2022). https://doi.org/10.1007/s42823-022-00404-z

5.      In the conclusion section, the author claimed that Ru its presence was welcomed in the bimetallic catalysts diminishing the size of the metal particles bimetal catalyst. The reason is recommended to be discussed.

Author Response

Please find attached the responses to your comments

Reviewer 2 Report

Authors studied formic acid dehydrogenation over Ru and Pd catalyst on fixed bed reactor. Overall the authors showed some interesting results.

Having said that authors must consider the following points.

If they used isothermal conditions on the fixed bed, they  must show some temperature profiles measurements along the reactor to prove that. Also reactor set-up is important to present in the manuscript.

Validation of the experimental results with a theoretical model is crucial to show the validity of the experimental findings.

Author Response

please find attached the responses to your comments

Reviewer 3 Report

This work is devoted to the investigation of formic acid decomposition over Ru and Pd-based catalysts both in liquid and gaseous phase. The manuscript is of general interest for the readers, it is scientifically rigorous and well structured. In my opinion it can be accepted after that the following points will be addressed

·       Please add in the introduction previous works on formic acid decomposition especially in the liquid phase environment to guarantee a wider state of the art for the readers (10.1016/j.cej.2018.12.137, 10.1016/j.cherd.2022.03.048)

·       Please explain the mineralization technique used prior to ICP-OES in the methodology section

·       The authors should explain if possible internal mass transfer limitations were assessed

·       Avoid the question in row 251, since it is not formally well accepted

Author Response

please find attached the response to your comments

Round 2

Reviewer 2 Report

Authors should do a preliminary validation of their experimental data to prove validity.
